# Effects of the Structure and Operating Parameters on the Performance of an Electric Scooter

**Le Trong Hieu and Ock Taeck Lim \***

School of Mechanical Engineering, University of Ulsan, Ulsan 44610, Republic of Korea; letronghieu1411177@gmail.com
\* Correspondence: otlim@ulsan.ac.kr; Tel.: +82-10-7151-8218

**Abstract:** The research objective is to approach the dynamic and consumed electrical energy of an electric scooter by varying the key input parameters, including rider mass, electric scooter mass, wind speed, wheel radius, and slope grade. A simulation model of an electric scooter was applied in a MATLAB-Simulink environment to investigate the scooter velocity, required power, battery voltage, and propulsion torque of the e-scooter. It was established by employing mathematical equations during the of electric scooters. The study found that the scooter velocity and electricity consumption were optimized by 3.9% and 0.08%, respectively, when the scooter weight decreased from 26 to 10 kg. The scooter velocity, electricity consumption, and required power decreased by 23.2%, 0.55%, and 8.56%, respectively, when the slope grade decreased from 1.15% to 0%. Following a wind speed reduction from 4 to 0 m/s, the consumed electricity and required power were optimized by 0.2% and 5.5%, respectively. The consumed electricity increased by 0.2% and the scooter velocity and required power significantly increased by 36.5% and 34.3% when the wheel radius increased from 0.105 to 0.185 m. Furthermore, the e-scooter could achieve an effective performance with a weight of 10 kg, wheel radius of 0.185 m, wind speed of 0 km/h, slope grade of 0%, and minimal rider weight. The simulation results showed that the scooter's effective performance range and consumed electrical energy could be optimized by suitably adjusting the key structures and operating parameters. To support this research, a concurrent experiment investigated the dynamic characteristics and electricity consumption of the electric scooter during operation. The experimental and simulated results had the same patterns in similar initial conditions.

**Keywords:** electric scooter; MATLAB-Simulink; scooter dynamic model; electric consumption

## 1. Introduction

Electric scooter are increasingly popular because of their comparatively low environmental pollution when traveling for short and medium distances [1,2]. In 2017, the e-scooter was first introduced as a new methodology of micromobility in the United States [3]. By the end of 2018, the e-scooter had overtaken the bicycle as the preferred vehicle, having an additional two million total trips [4]. As a stable form of transportation, e-scooters have the potential to change urban transport systems by reducing traffic congestion and being powered by electricity [5,6]. The e-scooter is especially useful in heavy traffic and where parking spaces are limited [7]. In addition, as compared to other transportation modes, the cost per kilometer traveled in electric scooters is relatively low [8,9]. within addition, the environmental impact of urban transportation can be reduced with the benefit of an electric scooter-sharing service [10].

From these above analyses, research involving electric scooters has continued. Zhejing Cao et al. [11] surveyed electric scooter users and evaluated a mixed logit model to investigate factors affecting the choice of electric scooters and transit in the Central Area of Singapore. The research results showed that the distance of transit transfer and access/egress walking decreases the share of a transit mode, and electric scooters are a great

alternative transport option to decrease mass rapid transit crowding in Singapore's Central Area. The electric scooter was more competitive than mass rapid transit network when the network indirectness was 2.276, there was at least one transfer, and 0.6 km longer walking distance. Through the survey, the research found that longer access/egress walking, more transfers, and higher levels of transit indirectly led to an increasing in the probability of using electric scooters. Andreas Nikiforiadis et al. [12] conducted a survey in Thessaloniki, Greece, with 578 questionnaires to understand the operation of e-scooters and the profile of their users, as well as their impact on transportation systems. The study indicated that males were more likely to use e-scooters, possibly due to a combination of risk-taking behavior and fascination with the novelty of e-scooters. Additionally, residents living in downtown areas reported higher regular usage compared with those living farther away from the city center. Tiziana Campisi et al. [13] also implemented a face-to-face questionnaire survey to examine the usage of personal mobility vehicles in the historic center of Palermo, analyzing survey results to provide strategies related to socially and environmentally friendly transportation systems. Along with this, Nurten Akgün-Tanbay et al. [14] conducted a face-to-face survey of 200 participants who used walking, cycling, and micromobility as their travel modes to investigate the impacts of perception of infrastructure, sociodemographic characteristics, frequency of road use, and road user perception on safety, comfort, and chaos in shared spaces. The findings highlight the importance of infrastructure and its impact on road users' safety, comfort, and chaos perceptions. Hongtai Yang et al. [15] conducted an analysis of the effect of electric scooter sharing on bike sharing in Chicago over 30 weeks; the analysis results showed that the overall bike sharing usage decreased by 10.2% per week per station with the appearance of electric scooter sharing. In addition, the volume of bike sharing decreased to 7.5%, 9.6%, and 20.5% for short, medium, and long-duration trips, respectively. In addition, Semih Severengiz et al. [16] presented a life cycle assessment methodology to assess the environmental effect of electric scooter sharing services under the aspect of ecological sustainability; the study shows that the global warming potential would be reduced by upgrading the battery with innovation in battery technology. The research indicated that the environmental impact of electric scooter sharing could be reduced by 4% by predictive maintenance. Recently, advanced battery packs and chargers have also expanded the scope of research and are widely used in electric vehicles [17,18]. Aree Wangsupphaphol et al. [19] investigated the life loss of both new and used batteries of retired electric vehicles for stationary energy storage in residential households and analyzed the technical and economic feasibility. The study demonstrated the advantage of using second-life batteries with a depth of discharge of 85%. Furthermore, the techno-economic analysis revealed that using the largest state of charge band and the most affordable second-life batteries (under USD 50/kWh) offered the most favorable payback period, approximately half of the project's 10-year lifespan. Himadry Shekhar Das et al. [20] developed a large-scale electric vehicle grid integration system and demonstrated the implementation of vector control in a grid-connected inverter to support the power grid. By leveraging proper control technology, the EVGI system can act as a grid energy storage system and provide services such as real and reactive power compensation, voltage and frequency support, and harmonics improvement. The study demonstrates the potential of EV charging stations to contribute valuable resources to the future grid by providing grid ancillary services. Besides the studies related to environmental benefits, convenience aspects, and factors affecting the sharing of electric scooters, there were other studies which studied electric scooter efficiency. The research of Byeong-Mun Song et al. [21] focused on the high-efficiency development of electric motorcycles with external rotor-type permanent-magnet (PM) motors. The team designed, built, tested, and integrated an outboard rotor-type PM motor onto the wheel rim. The motor was supplied with 48 V dc power from four 12 V batteries and pulse width modulated (PWM) inverter voltage. The research results showed that the all-in-all efficiency was 87% at the rated speed of 30 km/s. When input voltages of 48 V and 54 V were set, the torque capability efficiently enabled the motor to drive the electric vehicle at all ranges of speed. In addition,

Yee-Pien Yang et al. [22] presented an energy management system with a regenerative brake and electronic gearshift system for an electric scooter; the efficiency from battery to wheel was more than 70% and the electric scooter gross efficiency with regenerative braking was optimized by 20%. Along with this, previous studies also studied electric scooter performance. M.N. Yuniarto et al. [23] explored the design and performance of the bBrushless direct current motor controller. There were six permutation steps in a proportional-integral-derivative current control controller. The research team used the PID control method in two control stages. When the test was performed, the motor rotated until it reached its maximum rotational speed with no load, then new loads were added in stages until the motor spun without being able to resist the load of the dynamometer. From the research results, it was shown that the controller could operate immediately at a capacity of 5000 watts with a torque of 60 Nm at 1000 rpm. In addition, Muhammad Nur Yuniarto et al. [24] conducted research to predict the behavior and characteristic of electric motorcycles based on simulation models and experimental tests. The research result showed that the errors for predicting energy consumption were 8.63% and 4.7% when the experimental tests were conducted with a dynamometer and normal on-road testing, respectively. However, detailed research on the influence of key factors on energy consumption and the performance of electric scooters was not carried out.

In summary, most of the previous research has focused on analyzing the environmental impact, flexible mobility value in urban area transport, traffic reduction ability in peak hours, and competition of electric scooters with traditional transport, as shown in Table 1. Some studies have been carried out to investigate electric scooter performance, such as innovative permanent magnet motor design, applied energy management system and regenerative braking, and using the PID method to control the power and torque of brushless direct current motor with the aim of improving electric scooter performance, but it has not been studied comprehensively. In particular, the effective performance of electric scooters under the effect of structural and operating parameters is a novelty aspect that can improve scooter dynamics and consumed energy during operation. In addition, the timing to achieving a stable velocity and voltage related to operating behavior is among the most interesting aspects yet to be thoroughly explored. This is a significant gap in electric scooter research, and is a motivation for this paper: to conduct research on the effects of structure and operating parameters on electric scooters. This paper will comprehensively examine the impact of key factors on performance and consumed energy of electric scooters. The electric scooter operation is simulated by dynamic operation equations based on certain operating conditions. The dynamic equation and control parameter was solved with MATLAB-Simulink [25]. From the achieved simulated results, the consumed energy and scooter dynamics can be optimized by adjusting suitable key parameters, including ES mass, slope grade, wheel radius, and wind speed. Furthermore, the timing for the ES to reach stable velocity was also analyzed under the influence of key factors. To support simulation results, an experimental test was conducted in the same conditions to evaluate the performance characteristics of an electric scooter. LabVIEW software was used to record experimental data.

**Table 1.** Comparison of conducted work and related work in previous studies.

| Reference | Proposed | Finding | Limitation |
|---|---|---|---|
| Zhejing Cao et al., 2021 [11] | Survey on electric scooter users in the Central Area of Singapore. | Longer access/egress walking, more transfer, and a higher level of transit indirectly led to increasing probability of use of electric scooters. | The study focused specifically on the Central Area Singapore and may not be fully representative of other areas. |
| Andreas Nikiforiadis et al., 2021 [12] | Survey of 307 e-scooter users to understand the operation of e-scooters and the profile of their users. | Shared e-scooters primarily replaced walking and public transport trips. Gender affected e-scooter use. | The research was conducted only in Thessaloniki, Greece. |
| Nurten Akgün -Tanbay et al., 2022 [14] | Survey of 200 participants on influence of perception of infrastructure, frequency of road use, and road user perception on safety, comfort, and chaos in shared spaces. | A higher perception of infrastructure positively impacted the safety and comfort perceptions of both walking and cycling. | The sample size was relatively small. The study does not investigate actual safety incidents or objective measures of comfort and chaos in shared spaces. |
| Hongtai Yang et al., 2021 [15] | Analysis of electric scooter sharing on bike sharing in Chicago over a period of 30 weeks. | The introduction of electric scooter sharing had a negative impact on bike sharing. | The analysis was conducted over a relatively short period of 30 weeks. The study didn't examine the satisfaction or experiences of e-scooter users or bike sharing users. |
| Semih Severengiz et al., 2021 [16] | A life cycle assessment methodology to evaluate the environmental impact of electric sharing services with a focus on ecological sustainability. | Upgrading the battery technology of electric scooters can lead to a reduction in global warming potential. | Other dimensions of sustainability, such as social and economic aspects, were not considered. |
| Aree Wangsupphaphol et al., 2023 [19] | Examining the use of retired electric vehicle batteries for stationary energy storage in residential households. | Using a retired battery with a depth of discharge of 85% for stationary energy storage offers advantages over using a new battery with similar characteristics. | Lacked practical implementation to address these challenges and assess their feasibility. |
| Muhammad Nur Yuniarto et al., 2022 [24] | Developing of an electric scooter model to simulate its performance, particularly focusing on range estimation. | The developed electric scooter model is valid and has a satisfactory level of accuracy in predicting energy consumption by reducing 89% energy cost compared with internal combustion engines in a ride-hailing application. | The study focuses primarily on the energy consumption model of the electric scooter and its performance comparison with internal combustion engine scooters. |

## 2. Methodology

### 2.1. Experimental Setup

Figure 1 presents the components of the ES experimental setup. In order to collect and store experimental data, a LabVIEW program was installed on a local PC [26]. The ES energy was supplied through a Li-ion battery 36 V, 9 Ah to the dc motor and, in order to measure the ES's velocity and travel distance, an electric control unit (ECU) was installed on the ES frame, which read the signals from a photosensor. The ECU was connected to the PC to store the data when the experiment was conducted. The analog signals of the battery voltage were measured by an NI 9221 with a 60 V input channel at 800 kS/s, which was connected to the PC by the LabVIEW program, and the photosensor was utilized to measure the speed of the front wheel. A dc motor located at the rear wheel helped the ES to overcome friction, wind, and grade resistance force by creating propulsion force.

The condition of the experimental test was instituted by the following: the electric scooter weight of $M_{ES}$ = 18 kg, a rider weight of $M_D$ = 77 kg, and a wheel radius of $R_W$ = 0.125 m. The 60-s experiment was conducted alongside the Taehwa River in South Korea, where the road grade was 0% and the wind speed was roughly 0 km/h.

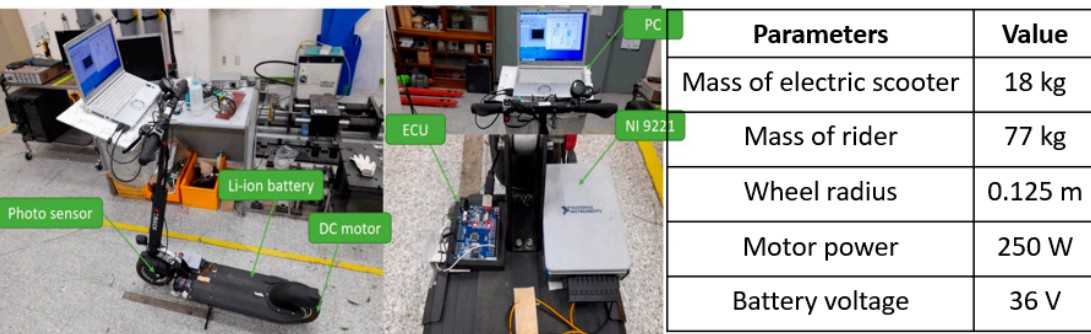

| Parameters | Value |
|:---|:---:|
| Mass of electric scooter | 18 kg |
| Mass of rider | 77 kg |
| Wheel radius | 0.125 m |
| Motor power | 250 W |
| Battery voltage | 36 V |

**Figure 1.** Experimental system and specifications of electric scooter.

### 2.2. Simulation Modeling

### A Dynamic Model of an Electric Scooter

A free-body diagram of an ES to simulate its dynamics during operation is shown in Figure 2.

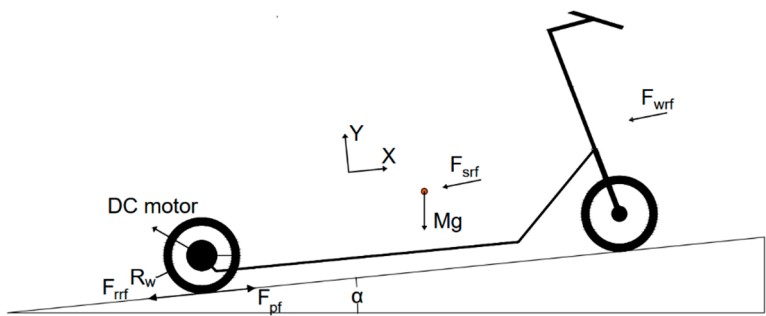

**Figure 2.** Model of an electric scooter.

The ES's motion follows Newton's second law, which can be described by:

$$F_{pf} - F_{srf} - F_{wrf} - F_{rrf} = Ma = M\frac{d^2x}{dt^2} \tag{1}$$

Based on Figure 2, the propulsion torque can be represented by following:

$$T_{pf} = F_{pf}R_w \tag{2}$$

where $F_{pf}$ represents the propulsion force; $T_{pf}$ is the propulsion torque;

$F_{wrf}$ represents the win resistance force, $F_{wrf} = \frac{1}{2}A_aC_a\rho(v_w + v_{ES})^2$;

$F_{rrf}$ represents the resistance force of rolling friction, $F_{rrf} = gMC_{rr}\cos\alpha$;

$F_{srf}$ represents the resistance force of grade, $F_{grf} = MgC_s$;

where $A_a$ is the frontal area, $C_a$ is the coefficient of aerodynamic drag, $\rho$ is the air density, $v_w$ is the wind speed, $v_{ES}$ is the scooter speed, M is the mass of rider and scooter, $C_{rr}$ is the rolling coefficient, g is the earth's gravitational force, and $C_s$ is the slope coefficient ($C_s = \sin(\alpha)$).

A dc motor was attached to rear wheel of the ES as shown in Figure 2. The propulsion torque can be recalculated by:

$$T_{pf} = (F_{grf} + F_{wrf} + F_{rrf} + M\frac{d^2x}{dt^2})R_w = T_m \tag{3}$$

where $T_m$ is the torque of dc motor, or

$$F_{pf}R_w = T_m \tag{4}$$

An ES employs a dc motor, which can be located on the front wheel, rear wheel, or both wheels. The dc motor creates propulsion torque to overcome friction, grade, and air resistance when an ES operates. The dc motor dynamic is modeled by Equations (5) and (6) [27,28].

$$E_c + R_a i_a(t) + L_a \frac{di_a}{dt} = U_a \tag{5}$$

$$B_1 w_m + j \frac{dw_m}{dt} = T_e - T_m = T_a \tag{6}$$

Armature current is the electrical variable of a dc motor, while speed is the mechanical variable of the dc motor. Based on Equations (5) and (6), it can be inferred that back EMF ($E_c$) and created torque ($T_e$) are proportional to the speed and armature current respectively, as presented by Equations (7) and (8).

$$E_c = K_b w_m \tag{7}$$

$$T_e = K_b i_a \tag{8}$$

The dc motor dynamic can be represented by substituting Equations (7) and (8) into Equations (5), and (6), and is represented by the following Equations (9) and (10):

$$L_a \frac{di_a}{dt} + i_a(t)R_a + K_b w_m = U_a \tag{9}$$

$$J \frac{dw_m}{dt} + B_1 w_m + T_m = K_b i_a(t) \tag{10}$$

where $L_a$ is the armature inductance, $i_a$ is the armature current, $R_a$ is the armature resistance, $K_b$ is the back emf constant, $w_m$ is the motor speed, $U_a$ is the dc motor voltage terminal, $J$ is the inertia torque, $B_1$ is the coefficient of viscous friction, and $T_m$ is the motor torque.

The propulsion force of the electric scooter was calculated by substituting Equation (4) into Equations (9) and (10); it was represented by the following:

$$i_a(t)R_a = U_a - L_a \frac{di_a}{dt} - K_b w_m \tag{11}$$

$$J \frac{dw_m}{dt} + B_1 w_m + F_{pf}R_w = K_b i_a(t) \tag{12}$$

By combining Equations (11) and (12), the propulsion force can be compacted and presented by the following equation:

$$F_{pf} = \frac{1}{R_w}\left[ \frac{K_b}{R_a}u - \frac{K_b L_a}{R_a}\frac{di_a}{dt} - J\frac{dw_m}{dt} - \left( B_1 + \frac{K_a K_b}{R_m} \right)w_m \right] \tag{13}$$

The propulsion torque can be rewritten as the following equation:

$$T_{pf} = \left[ \frac{K_b}{R_a}u - \frac{K_b L_a}{R_a}\frac{di_a}{dt} - J\frac{dw_m}{dt} - \left( B_1 + \frac{K_a K_b}{R_m} \right)w_m \right] \tag{14}$$

### Battery Model

The battery was installed on the ES and was used to supply the voltage for the dc motor. The Li-ion battery used in this research is represented by a discharge battery model. It can correctly describe the dynamics of voltage when the current varies, taking into account the OCV (open-circuit voltage) [29]. A term concerning the polarization voltage

is included to model the behavior of open-circuit voltage and, at the same time, the term concerning the polarization resistance was adjusted [30]. The battery voltage description is illustrated by the following Equation (15):

$$V_b = E_0 - \frac{KQ}{Q - i(t)}i(t) - R.i + Ae^{(-B.i(t))} - \frac{KQ}{Q - i(t)}i^* \tag{15}$$

where $V_b$ is the voltage of battery, $E_0$ is the voltage constant of battery, K is the polarisation constant, Q is the capacity of battery, i(t) is the actual battery charge, A is the exponential zone amplitude, B is the exponential zone time constant inverse, R is the internal resistance, i is the battery current, and i* is the filtered current.

### Scooter Performance

The total power employed to push a scooter and rider primarily overcomes friction, slope, and wind resistance.

$$P_{total} = P_{friction} + P_{grade} + P_{wind} \tag{16}$$

The power to overcome the resistance of wind, $P_{wind}$, can be presented by the following equation:

$$P_{wind} = \frac{1}{2}A_a C_a \rho (v_w + v_{ES})^2 v_{ES} \tag{17}$$

where $A_a$ is the frontal area, $C_a$ is the coefficient of aerodynamic drag, $\rho$ is the air density, $v_w$ is the wind speed, and $v_{ES}$ is the scooter velocity. The power to overcome a grade, $P_{grade}$, can be presented by the following equation:

$$P_{grade} = MgC_s v_{ES} \tag{18}$$

where M is the mass of rider and scooter, and $C_s$ is the slope coefficient ($C_s = \sin(\alpha)$).

The power to overcome friction, $P_{friction}$, can be presented by the following equation:

$$P_{friction} = gMC_{rr}\cos\alpha v_{ES} \tag{19}$$

where $C_{rr}$ is the rolling coefficient.

The electric motor power can be chosen for electric scooter during operation based on the consumed power. The total resistance power will be the sum of wind resistance power, grade resistance power, and friction resistance power in case the slope ratio is greater than zero. In this case, the required power for the dc motor must be greater than the total resistance power. Otherwise, the total resistance power includes the sum of wind resistance power and friction resistance power, and the required power for the dc motor must also be greater than the total resistance power, as shown in Figure 3.

### PID Controller for Controlling Scooter Speed

To control scooter speed, a control loop feedback mechanism widely applied in industrial control systems, the proportional integral derivative (PID), was used. The PID controller was applied to continuously calculate an error e(t), the difference between a desired point and a variable [31,32]. The PID controller can be presented by the following equation:

$$U(t) = K_p e(t) + K_i \int_0^1 e(\tau)d\tau + K_d \frac{de(t)}{dt} \tag{20}$$

where $K_p$ is the proportional gain, $K_t$ is the integral gain, $K_d$ is the derivative gain, U(t) is the control variable, and e(t) is the control error.

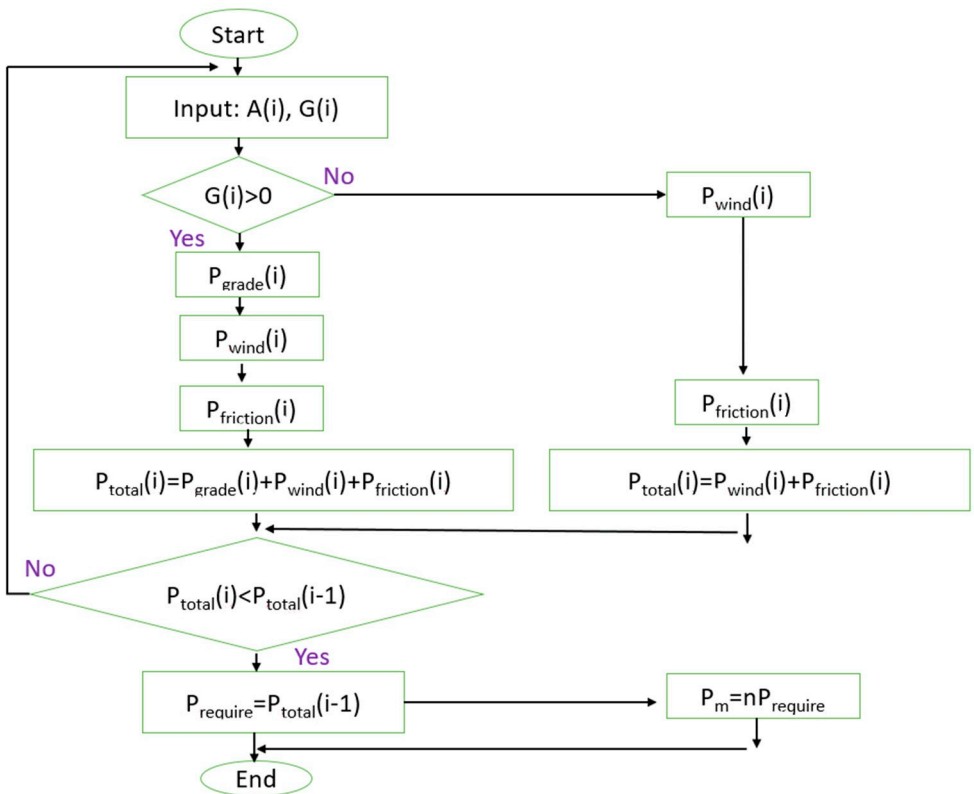

**Figure 3.** Flowchart to calculate the power of electric scooter.

The speed-controlling algorithm of the electric scooter using the PID controller is shown in Figure 4. The PID controller used in this paper considered the voltage as a control variable, which was adjusted to control the total torque and the motor torque on a scooter wheel ($T_{pf} + T_m$).

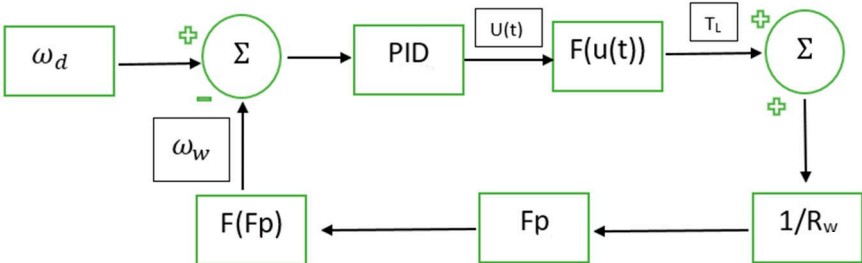

**Figure 4.** Schematic of scooter speed control using a PID controller.

## 3. Results and Discussion

### 3.1. Model Validation

The simulation and experiment were implemented under initial conditions of rider mass, wheel radius, slope ratio, and wind speed of 77 kg, 0.125 m, 0%, 0 km/h, respectively, while the scooter weight was varied from 26 kg to 18 kg. Figure 5a,b show the influence of scooter mass on velocity and travel distance. The same trend can be observed from the simulation and experimental data values. In both cases, the velocity and travel distance were reduced when the ES weight was raised from 18 kg to 26 kg. The main cause was that the increase in scooter mass led to an increase in the total resistance force while the other conditions remained constant. The maximum differences in the velocity of the ES of weights 18 kg and 26 kg were 9.41% at 45 s and 7.72% at 35 s, respectively. The maximum differences in the moving distance of the ES of weights 18 kg and 26 kg were 1.63% and

3.08%, respectively. This is acceptable in terms of maximum scale as such velocity and moving distance values are average values.

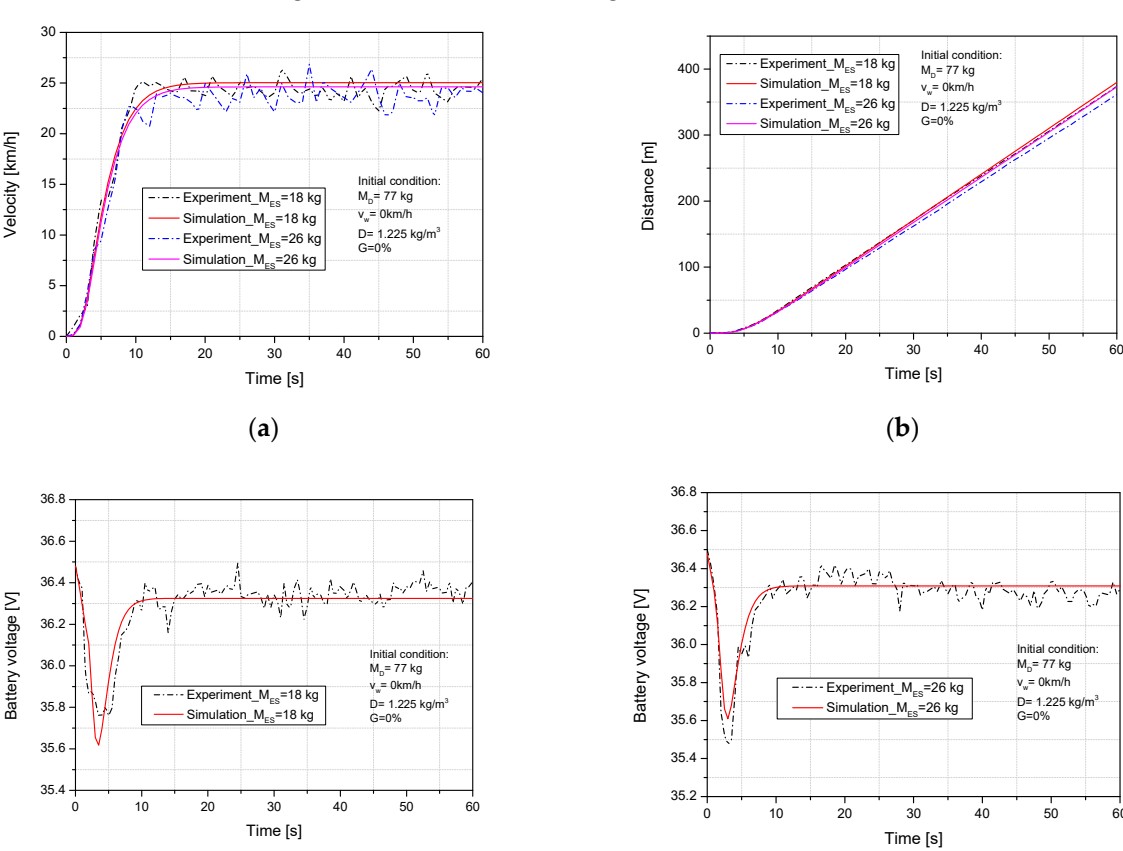

**Figure 5.** Impact of scooter weight on (**a**) velocity, (**b**) travel distance, (**c**) and (**d**) battery voltage.

Figure 5c,d compare the experimental and simulation battery voltage when reducing the scooter weight from 26 kg to 18 kg. The electricity consumption was reduced in both the simulation and the experiment. The reduction of electric consumption is explained by reducing the load on the motor and thus the energy demand. The maximum differences are 0.38% and 0.47%, corresponding to scooter weights of 26 kg and 18 kg, respectively. In the experiment, the battery voltage value is considered an average value, therefore, it is acceptable for the maximum scale as a result. To decrease the velocity and battery voltage difference between the experimental and simulation results, aside from input corrections in view of rider mass, slope ratio, and wheel radius, sudden wind should be controlled to reduce that error.

One of the parameters that is difficult to avoid during ES operation is the rider mass, which directly affects the dynamic characteristics of the ES during movement. This simulation and experiment were run to examine the impacts of driver mass on the characteristics of motion with initial conditions of 0 km/h wind speed, 0% slope grade, 0.125 m wheel radius, and 18 kg ES mass when the rider weight was raised from 77 kg to 87 kg. The results are shown in Figure 6. We can observe that the velocity and travel distance from the experiment and simulation were decreased when the rider weight was increased from 77 kg to 87 kg, as shown in Figure 6a,b. The maximum differences between the experiment and simulation of velocity were 9.59% at 24 s and 9.21% at 47 s when the weight of the rider was 77 kg and 87 kg, respectively. Figure 6c,d compares the effects of driver mass on battery voltage during the simulated and experimental ES operation. From the simulation results, the consumed electricity was decreased by 0.06% when the rider weight was changed from 87 kg to 77 kg, and the timing of battery voltage stabilization was shortened from 14.9 s to

14.4 s in the same conditions. The reason for this decline in electricity consumption was the decrease in total resistance force that led to reducing the load on the motor. The maximum differences were 0.4% at 28.5 s and 0.39% at 46.5 s with a rider weight of 87 kg and 77 kg, respectively. This maximum scale is tolerable because the battery voltage is considered another average value during the experiment. As with the velocity discrepancy described above, the battery voltage discrepancy between the experiment and the simulation was caused by a sudden wind in the actual road test. The velocity error between the simulation and experiment was caused by a sudden appearance of wind during the experiment.

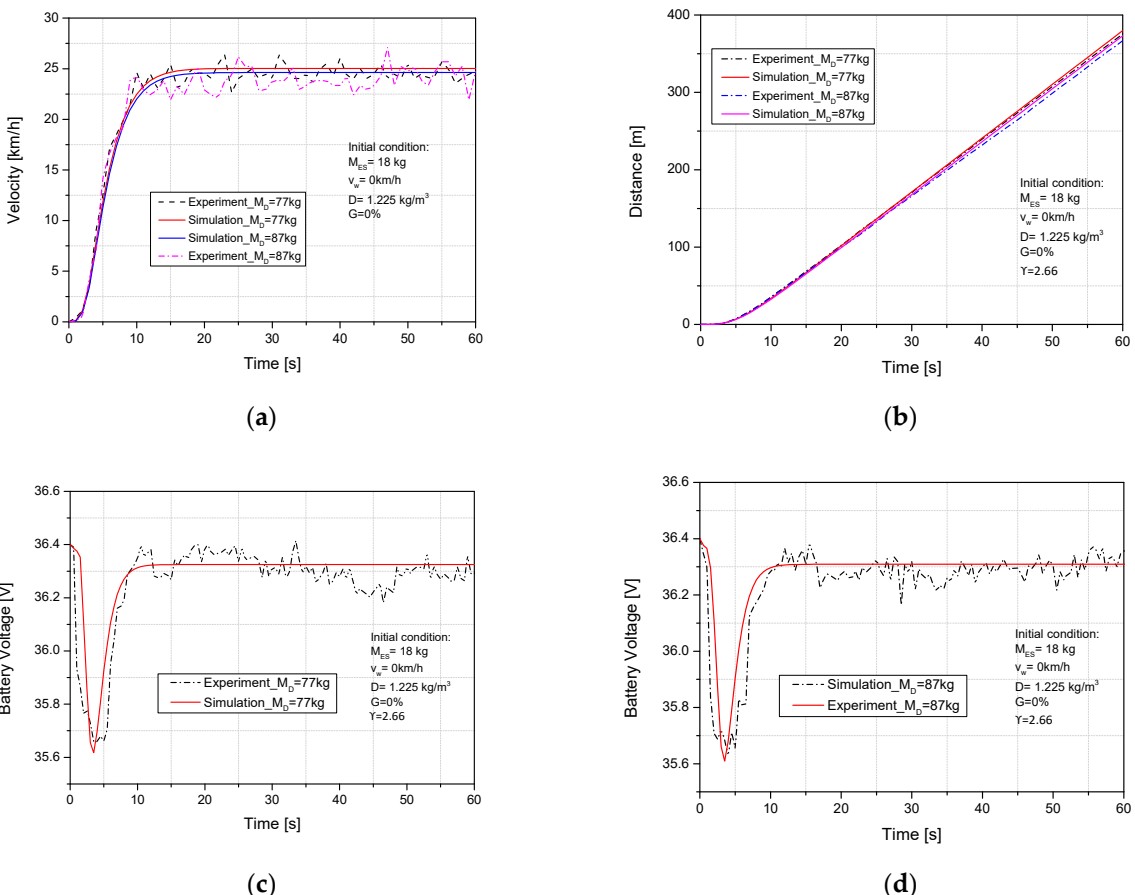

**Figure 6.** Impact of rider weight on (**a**) velocity, (**b**) distance, (**c**) and (**d**) battery voltage.

### 3.2. Effect of Operating and Structure Parameters on Electric Scooter Performance

One of the important factors that impacts the operating behavior of an electric scooter is its mass, which directly effects an electric scooter's performance because of its relationship with friction and grade resistance. Our simulation was run under the initial conditions of rider weight of 77 kg, wheel radius of 0.125 m, slope ratio of 0%, and wind speed of 0 km/h. The ES mass was varied from 10 kg to 26 kg to examine the dynamics and electricity consumption of the ES. We can observe that the velocity maximum increased from 24.48 km/h to 25.44 km/h (3.9%) when the scooter mass reduced from 26 kg to 10 kg. In addition, the timing for the ES to reach stable speed decreased from 14.3 s to 13.7 s, as shown in Figure 7a,b, showing that required power decreased from 515 w to 505.5 w in the same conditions. Furthermore, the electric consumption decreased 0.08% percent in stable voltage period by decreasing the scooter mass from 26 kg to 10 kg, as shown in Figure 7c. The max propulsion torque was decreased from 28.8 N.m to 28.5 N.m in same conditions, as presented in Figure 7d. The decrease in scooter mass caused an increase in scooter velocity and a decrease in required power. Because the friction resistance force decreased with the decrease in scooter mass while the initial conditions were kept a constant, the total resistance force was decreased, which led to a decrease in the load on the dc motor.

This also explains the reason for the decrease in electric consumption. This implies that the dynamic and consumed electric characteristics during operation can be optimized by decreasing the scooter mass. From the simulation results, an electric scooter mass of 10 kg is a potential parameter for the scooter reach an effective performance area.

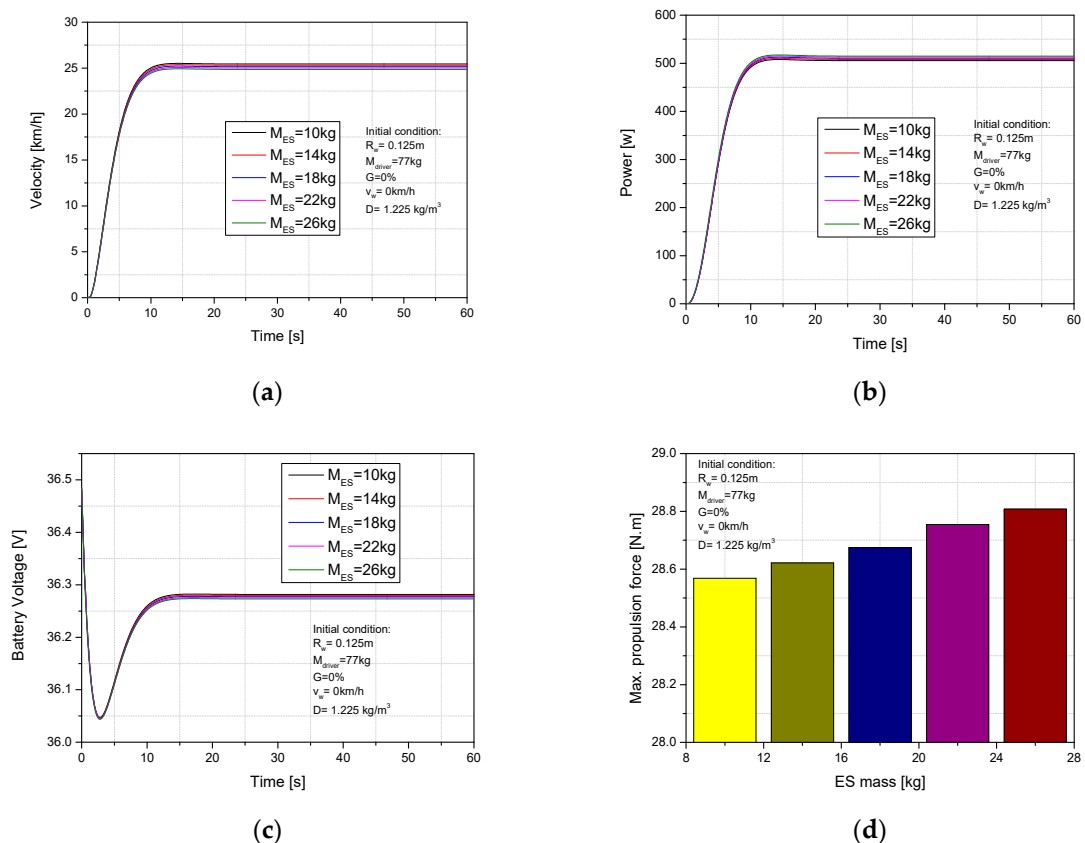

**Figure 7.** Impact of the ES mass on (**a**) velocity, (**b**) required power, (**c**) battery voltage, (**d**) propulsion torque.

One of the operational parameters directly influencing the dynamic and consumed electricity characteristics of electric scooters is rider mass. Because it relates to the friction and slope resistance force, to understand how rider mass affects the performance and consumed electricity characteristics of an electric scooter, the simulation was run for 60 s under these initial conditions: scooter weight of 18 kg, wheel radius of 0.125 m, wind speed of 0 km/h, and slope ratio of 0%. Rider mass has a sensitive influence on performance and electricity consumption. As is observed in Figure 8a,b, the velocity and required power were significantly changed from 19.57 km/h to 24.6 km/h and from 555 w to 518 W, respectively, when the rider mass was decreased from 150 kg to 70 kg. Figure 8c,d show the effect of rider mass on battery voltage and propulsion torque of the ES during operation. The propulsion torque decreased 31 N.m to 28.8 N.m, and the consumed electricity was decreased 0.19% when the driver mass was reduced from 150 kg to 70 kg. The battery voltage stabilization timing was shortened from 14.7 sec to 14.1 sec in the same conditions. The rise of velocity as well as the reducing power could be disclosed by the decrease of rider mass leading to decrease of the friction resistance force while remaining the other conditions. It also is a reason for reducing electric consumption. The total resistance force was decreased by changing of rider mass from 150 kg to 70 kg. It led to decreasing load on the dc motor therefore the request energy supplies for the dc motor was reduced. From the simulation results, an performance and consumed energy of electric scooter gets an effective area with the decreasing of rider mass. This implies that dynamic and electric consumption optimization depends on a great deal driver mass.

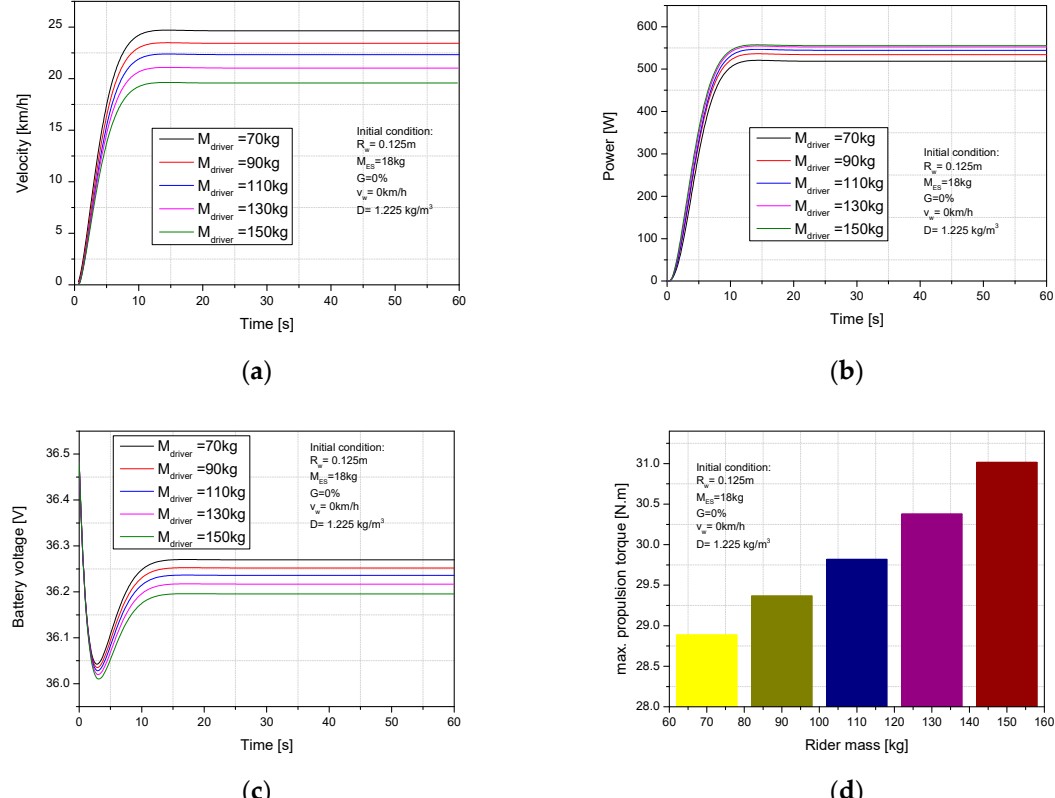

**Figure 8.** Impact of the driver mass on (**a**) ES velocity, (**b**) require power, (**c**) battery voltage, (**d**) propulsion torque.

The impacts of terrain are difficult to avoid when driving an ES under various operating conditions. Therefore, the effect of slope ratio was studied to investigate how the road slope grade affected the ES performance and consumed electricity. Figure 9a,b show the influence of slope grade on velocity as well as the ES's required power when the slope ratio was varied from 0% to 1.15%. The velocity and required power were significantly changed from 25.15 km/h to 20.4 km/h (23.2%) and from 510.6 w to 554.1 w (8.56%), respectively. Furthermore, the time to reach stable behavior increased from 14.7 s to 15.4 s. The effects of slope grade on consumed electricity and propulsion torque of electric scooters are shown in Figure 9c,d. It can be observed that the max propulsion torque increases from 28.6 N.m to 30.6 N.m (6.9%), corresponding to the slope ratio being adjusted from 0% to 1.15%. Moreover, the consumption of electricity increased by 0.55% under the same condition. The increased slope grade caused a significant increase in slope resistance force along with a slightly reduced rolling resistance force, so the total resistance force was significantly raised. It led to an increased load on the motor, so the required energy for the motor increased. This explains why the required power and electric consumption increased, as well as the reduction in velocity.

Weather conditions are difficult to avoid when using an ES. Therefore, wind needs to be fully considered to see how it affects the performance and consumed electric energy of the electric scooter. The impacts of wind speed on velocity, required power, electric consumption and propulsion torque are presented in Figure 10a,d. The wind velocity was raised from 0 m/s to 4 m/s while the other conditions consisted of rider mass, electric scooter mass, wheel radius, and slope grade of 77 kg, 18 kg, 0.125 m, and 0%, respectively. The velocity significantly dropped from 25.03 km/h to 23.05 km/h (8.5%) when the wind velocity was adjusted from 0 m/s to 4 m/s, which can be noted in Figure 10a. In addition, the times to reach stable behavior were 13.4 s and 14.2 s, respectively. Figure 10b presents the required power with rising wind speed. We can observe that the required power significantly rose from 510 w to 540.5 w (5.9%) with the rising of wind speed. The ES's

electric consumption rose 0.2% and the timing to the battery voltage stabilization extended from 12.7 s and 13.5 s when the velocity of wind was adjusted from 0 m/s to 4 m/s. In addition, the propulsion torque rose from 28.7 N.m to 29.6 N.m in the same conditions, as shown in Figure 10c,d. The increase in wind speed caused an increase of wind resistance while the other conditions were kept constant. It caused an increase in load on the motor and energy demand. As a result, the required power and electric consumption increased.

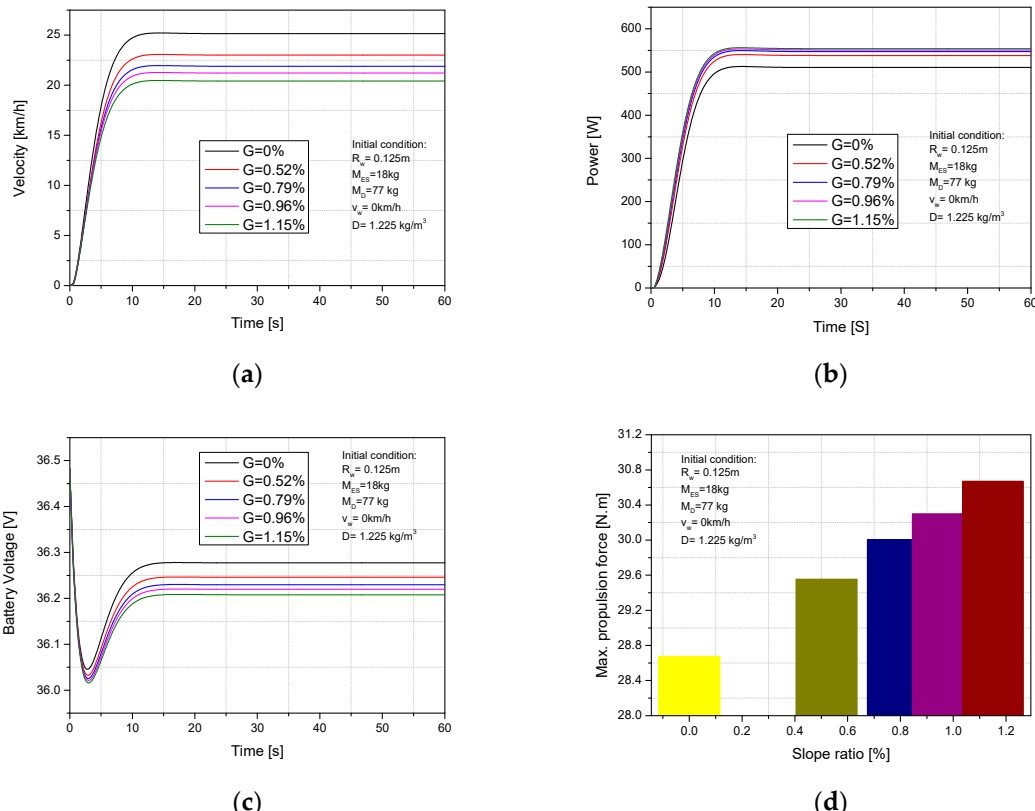

**Figure 9.** Impact of the slope grade on (**a**) ES velocity, (**b**) required power, (**c**) battery voltage, (**d**) propulsion torque.

One of the structural parameters that has an important impact on the motion of an ES is the radius of the wheel, as it is directly related to the rotational inertia of an ES during operation. The wheel radius was increased from 0.105 m to 0.185 m while the other parameters were kept constant ($M_R$ = 77 kg, $M_{ES}$ = 18 kg, G = 0%, $v_w$ = 0 km/h). Figure 11a shows that the velocity of the electric scooter significantly increased from 22.7 km/h to 31 km/h (36.5%) when the radius of the wheel was adjusted from 0.105 m to 0.185 m. with that change, the required power of the electric scooter increased from 373 w to 569 w (34.4%), as shown in Figure 11b. In addition, the increase of electric consumption was found to be roughly 0.2% by battery voltage reduction when the wheel radius was adjusted from 0.105 m to 0.185 m. Furthermore, the propulsion force increased from 28.4 N.m to 29.4 N.m in the same conditions, as shown in Figure 11c,d. The increase of wheel radius caused an increase in the load on the motor, so higher energy was required to obtain a certain velocity. This is why the required power and electric consumption increased. From the achieved results, the electric scooter reached an effective performance area with wheel radius $R_w$ = 0.185 m. The wheel radius ($R_w$ = 0.185 m) was chosen as a prospective input parameter for the next examination.

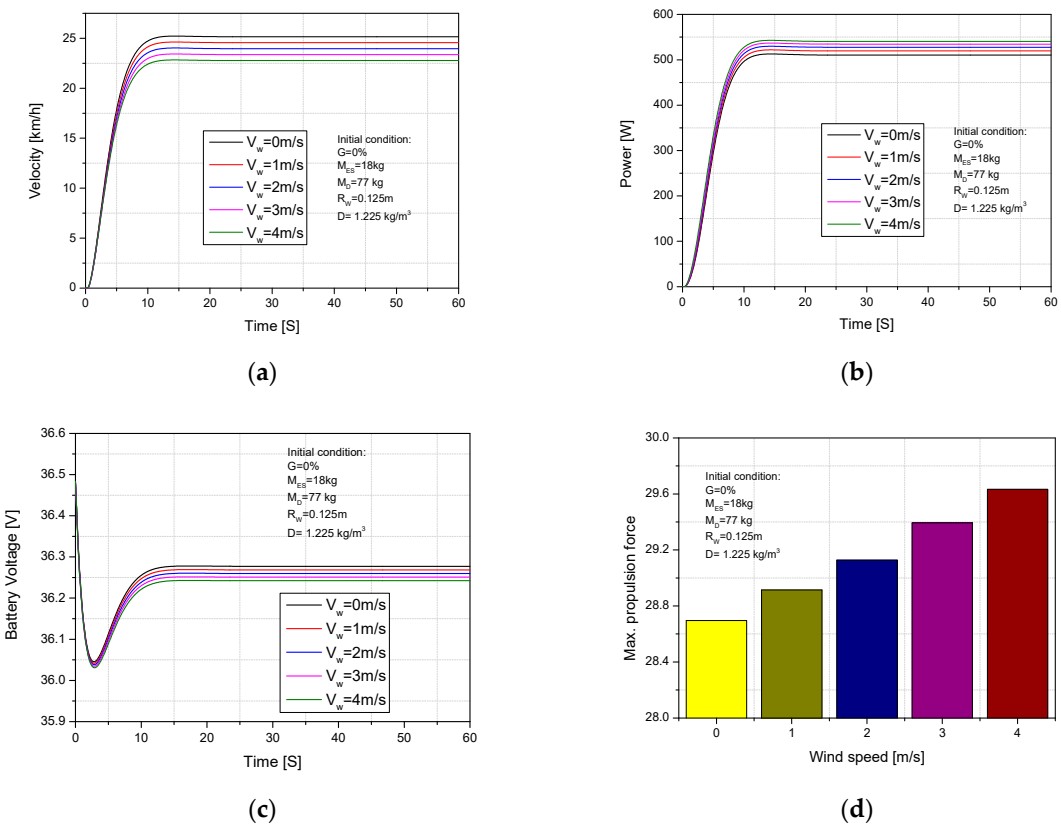

**Figure 10.** Impact of the wind speed on (**a**) ES velocity, (**b**) required power, (**c**) battery voltage, (**d**) propulsion torque.

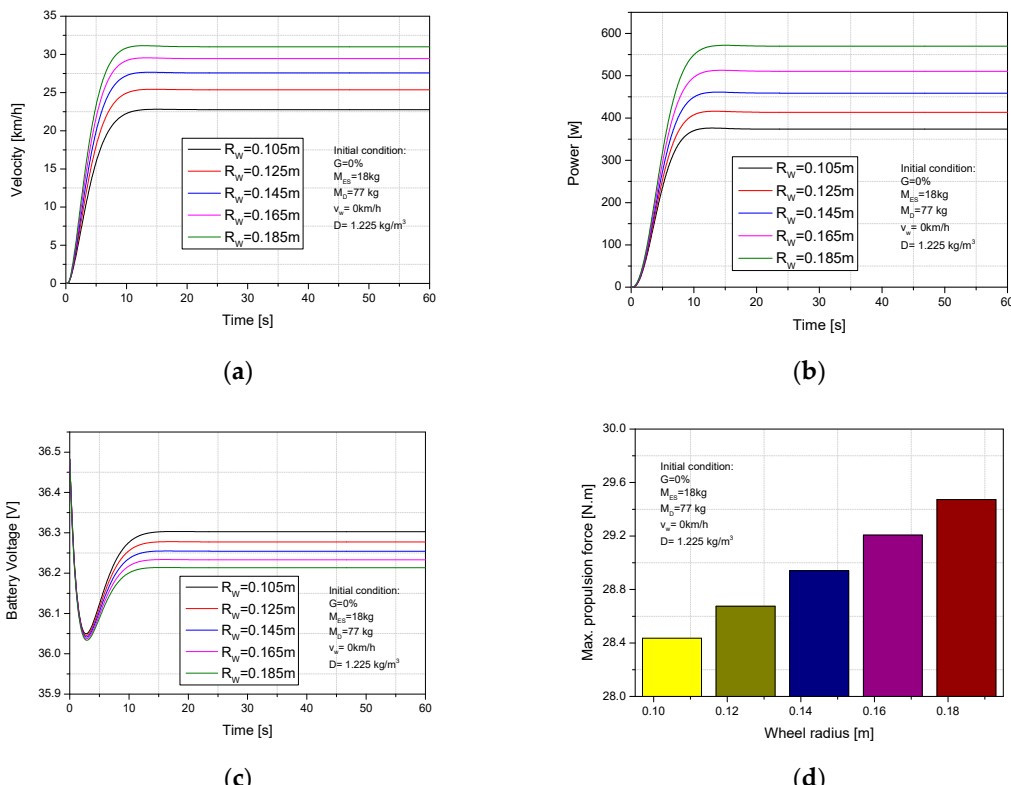

**Figure 11.** Impact of the wheek radius on (**a**) ES velocity, (**b**) required power, (**c**) battery voltage, (**d**) propulsion torque.

The frontal area is an important factorthat has a sensitive effect on the dynamics and battery voltage of electric scooters. The frontal area was changed from 0.4 to 1.2 m$^2$ while the other parameters were kept constant ($M_R$ = 77 kg, $M_{ES}$- = 18 kg, $R_w$ = 0.125 m, G = 0%, $v_w$ = 0 km/h). The velocity significantly decreased from 26.9 to 20.6 km/h when the frontal area was increased from 0.4 to 1.2 m$^2$ the required power increased from 476 to 553 W in the same conditions, as shown in Figure 12a,b. This can be explained by the frontal area increase leading to an increase in wind resistance; therefore, the total resistance force rose. Figure 12c shows that the battery voltage decreased from 36.3 to 36.21 V (0.26%). Furthermore, the propulsion force increased from 28.4 to 28.88 N.m in the same conditions, as shown in Figure 12d. The total force resistance force increasing led to an an increased energy demand to supply to the dc motor. This is why the electric consumption and propulsion torque increased when the frontal area increased.

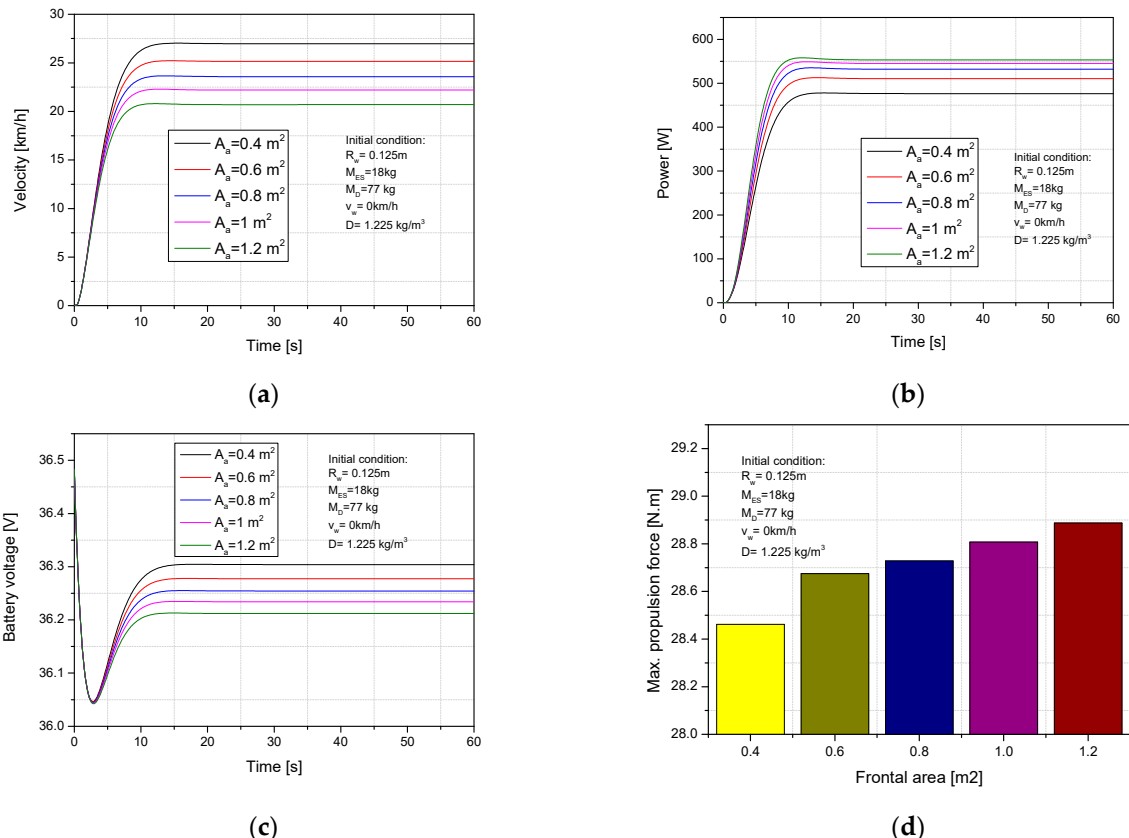

**Figure 12.** Impact of the frontal area on (**a**) ES velocity, (**b**) required power, (**c**) battery voltage, (**d**) propulsion torque.

### 3.3. Speed Control Results of the Electric Scooter

To approach the flexible behavior of electric scooter during operation, a model control study using a PID controller under the effects of wind speed and slope grade were implemented. Figure 13 presents the comparison of scooter speed under PID and non-PID control with an increase in slope grade of 1.15%. The results show that the scooter speed suddenly decreases and slowly approaches a steady speed with non-PID control. With PID control, the temporary velocity decrement has reduced after the slope grade is activated; after the decrement period, the scooter velocity quickly reaches a steady speed, as the result of propulsion torque control based on the adjusted activity voltage signal.

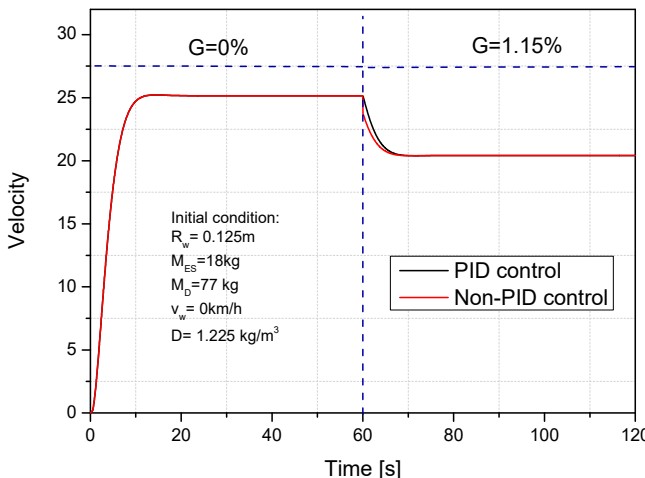

**Figure 13.** The comparison results of velocity between PID and Non-PID control with increasing slope grade 1.15%.

Figure 14 shows the effect of increasing slope grade to speed control under using a PID controller. The total time is 120 s. In the first stage, the slope grade was maintained at 0% within an initial 60 s. The slope grade was varied to 0.52%, 0.79%, 0.96% and 1.15% in the second stage. Figure 14a shows that the scooter velocity quickly achieved a stable value during the first stage of 60 s, when the slope grade was increased; the scooter velocity slowly decreased and rapidly reached a steady value in the second stage. Because the slope grade suddenly increased while the scooter velocity was relatively high, the real velocity was slow for a certain duration and the higher resistance when the slope grade increased was the cause of decreasing scooter velocity in the second stage. Figure 14b presents the required power with increasing slope grade while using PID control. The required power rapidly increased and reached a maximum value when the slope grade was increased, but it quickly dropped and obtained a certain value because of the controlled scooter speed. This can be explained by the higher slope grade being a cause for higher power to overcome the increasing resistance force.

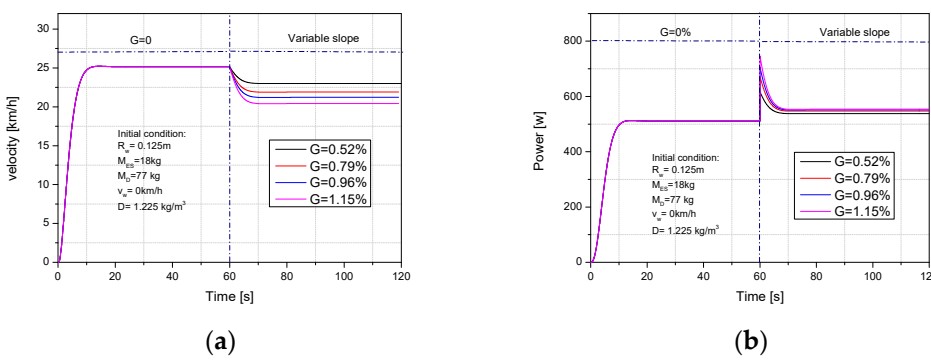

**Figure 14.** The velocity and required power of PID control with changing slope grade: (**a**) Velocity and (**b**) required power.

The effect of wind speed on scooter velocity and required power using PID control is shown in Figure 15a,b. Here, the wind speed was maintained at 0 km/h during 60 s of the first stage, then the wind speed was varied to 1, 2, 3, and 4 m/s in the second stage. As can be observed in Figure 15a, the scooter velocity rapidly reached a stable value in the first period. The scooter velocity slowly decreased because of the sudden increase in wind resistance while the velocity was in a high and stable period when the wind speed was activated, along with the voltage value being adjusted by PID controller. Thus, the scooter velocity slowly decreased and rapidly reached a stable value. Figure 15b shows that, when

the wind speed is activated in the second stage, the required power value rapidly increases and reach a maximum value and then slowly reduces, the power value also reaches a steady value because the scooter velocity is controlled.

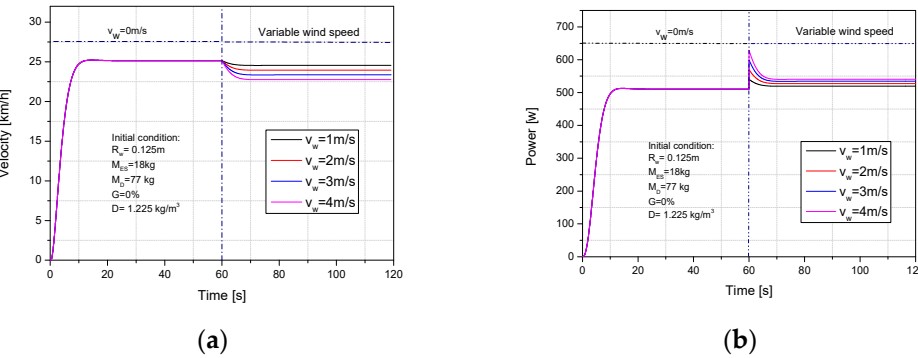

**Figure 15.** The velocity and required power of PID control with changing wind speed: (**a**) Velocity and (**b**) required power.

## 4. Conclusions

The electric scooter (ES) was mathematically modeled and simulated by MATLAB-Simulink through analysis of its operational dynamics. The influence of input parameters such as rider weight, scooter weight, wheel radius, slope grade, and wind speed on the total resistance force and on the amount of energy consumed during ES operation behavior was investigated.

The research results are summarized as follows: The velocity increased by 3.9% and the electric consumption was optimized by 0.08% when the scooter weight was decreased from 26 kg to 10 kg. The required power and consumed electricity were optimized by 8.56% and 0.55% by decreasing the slope grade from 1.15% to 0%. The consumed electric and required power were optimized by 0.2% and 5.5% when the wind speed was changed from 4 m/s to 0 m/s. The consumed electricity increased by 0.2% and the velocity and required power significantly increased by 36.5% and 34.4% by adjusting the wheel radius from 0.105 m to 0.185 m.

These research results indicate that the performance efficiency and electricity consumption of an ES can be easily optimized by suitably adjusting structured parameters and depending on considerable operation parameters. To support our MATLAB-Simulink ES simulation, the same ES was experimentally road tested alongside the Taehwa River in South Korea under the same input conditions to examine its operational dynamics and electricity consumption. The research only focuses on the optimization of the scooter's performance and energy consumption, without considering other aspects such as ride comfort and control of energy consumption during the start-up period. For future research on electric scooters, one of the interesting fields is the control-consumed energy of electric scooters. An input design and operation parameters were modeled, which affected the consumed electricity and velocity during the start-up period of the electric scooters, will be investigated to optimize the performance efficiency of an electric scooter. An experiment system will be implemented on electric scooters to validate the simulated results from the controlled model.

**Author Contributions:** L.T.H.: Conceptualization, Methodology, Investigation, Writing—Reviewing, Formal analysis, Resources, and Editing. O.T.L.: Supervision, Project administration. All authors have read and agreed to the published version of the manuscript.

**Funding:** This results was supported by "Regional Innovation Strategy (RIS)" through the National Research Foundation of Korea (NRF) funded by the Ministry of Education (MOE) (2021RIS-003). This work was supported by the Technology Innovation Program (or Industrial Strategic Technology Development Program—The Safety Based Technology Development and Substantiation of Small Hydrogen Powered Vessel) (RS-2022-00142947, The technology development on fuel cell electric

propulsion system using Land Based Test Site) funded By the Ministry of Trade, Industry & Energy (MOTIE, Korea).

**Institutional Review Board Statement:** Not applicable.

**Informed Consent Statement:** Not applicable.

**Data Availability Statement:** Not applicable.

**Acknowledgments:** The authors appreciate the anonymous reviewers for their constructive comments and suggestions that significantly improved the quality of this manuscript.

**Conflicts of Interest:** The authors declare no conflict of interest.

## Nomenclature

| | |
|---|---|
| $M$ | Total mass of electric scooter and driver, kg |
| $x$ | Moving distance, m |
| $A_a$ | Frontal area, m$^2$ |
| $C_a$ | Aerodynamic drag coefficient |
| $\rho$ | Air density, kg/m$^3$ |
| $v_w$ | Velocity of wind, km/h |
| $v_{ES}$ | Electric scooter velocity, km/h |
| $g$ | Gravitational of Earth, m/s$^2$ |
| $C_{rr}$ | Coefficient of rolling resistance |
| $t$ | Time, s |
| $R_w$ | Wheel radius, m |
| $V_b$ | Voltage of battery, V |
| $E_0$ | Battery constant voltage, V |
| $K$ | Polarization constant, V/Ah |
| $Q$ | Battery capacity, Ah |
| $i(t)$ | Actual battery charge |
| $R$ | Actual battery charge $i(t) = \int i\,dt$ |
| $i$ | Internal resistance |
| $i^*$ | Battery current, A |
| $A$ | Filtered current, A |
| $B$ | Exponential zone amplitude, V |
| $E_c$ | Exponential zone time constant inverse, Ah$^{-1}$ |
| $R_a$ | The back emf |
| $i_a$ | The armature resistance, $\Omega$ |
| $L_a$ | The armature current, A |
| $U_a$ | The armature inductance, H |
| $B_1$ | The terminal voltage of DC motor, V |
| $w_m$ | The viscous friction coefficient |
| $T_e$ | The speed of motor, rad/s |
| $T_m$ | The electromechanical torque, N.m |
| $T_a$ | The load torque, N.m |
| $J$ | The acceleration torque, N.m |
| $K_b$ | The torque of inertia |
| | The torque constant or back emf constant |

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
