# Peer review of "Effects of the Structure and Operating Parameters on the Performance of an Electric Scooter"

_sustainability, doi:10.3390/su15118976_

Round 1
Reviewer 1 Report
The paper titled "The effect of structure and operating parameters on the effective performance area of an electric scooter" is reviewed and is the good attempt by the researchers. However, the paper requires certain revisions prior to the acceptance.
1. The Literature section is fine and clearly presented.
2. Avoid the usage of words We/I/our etc. For eg: In line no-88, "From a brief summary of study in the area of electric scooter, we find that most of the".
3. In line No-91, the authors written that 'The studies have been done on small, light vehicles or devices 91 such as electric two-wheelers, but not studied comprehensively'. The authors must discuss the lapse that identified in the research of electric two-wheeler's so far. It adds more impact to the work.
4. It is recommended to add the significant contributions of this work at the end of the section-1.
5. Section-2 presented the mathematical modelling and is sufficient. however the authors are advised to verify whether all the parameters that used in the Equations are discussed or not. It is identified that, few of the parameters are missing in the description.
6. The results shown in Fig.8 are rendered by considering the driver mass up to 97Kg. The authors are advised to present the results by considering the driver mass along with the co-passenger mass (up to the weight of 150kgs).
Author Response
Dear Chief Editor and Reviewers
We would like to thank you very much for your kind consideration and facilitating the review of our manuscript. I’m submitting the detailed response with consideration of all comments and recommendations given by respective editor and reviewers. I believe that their comments helped the authors to improve the quality of the manuscript to a great amount, and in this regards I appreciate the valuable time they spent on reviewing the revise manuscript. Please note that the blue colored text is our responding to the reviewers, and the red colored text is the revised/added sections to the manuscript. Also, the added sections in the manuscript are colored in red.
Le Trong Hieu
Reviewer #1
Authors reply: Thank you very much for your valuable comments. The authors would like to response the reviewer comments below:
- The Literature section is fine and clearly presented.
The author would like to thank to the reviewer for the valuable’ comment. The paper was alos added some reference to improve the quality of paper. Changes are marked throughout the introduction section of the revised manuscript.
- Avoid the usage of words We/I/our etc. For eg: In line no-88, "From a brief summary of study in the area of electric scooter, we find that most of the".
The authors are very grateful for the comments and recommendations to improve the quality of this paper. The authors changed the word We/I/Our in line no-88. The authors already revised it in the revised manuscript.
- In line No-91, the authors written that 'The studies have been done on small, light vehicles or devices 91 such as electric two-wheelers, but not studied comprehensively'. The authors must discuss the lapse that identified in the research of electric two-wheeler's so far. It adds more impact to the work.
We would like to thank to the reviewers for the valuable comment and suggestions. The authors agree and accept the reviewer for the valuable comment and suggestions that the authors must discuss the lapse that identified in the research of electric two-wheeler's so far. It adds more impact to the work. Therefore, the authors have been made and revised the the section of introduction especially in the part that related to the lapse in research of electric two-wheels, namely electric scooter.
- It is recommended to add the significant contributions of this work at the end of the section-1.
We would like to thank the reviewer for the valuable comment and suggestion. Yes, the authors agree and accept the reviewer’ suggestions that the authors need add the significant contributions of this work at the end of the section-1. Therefore, the authors have been made a revision to the manuscript by add a contribution. The add parts were written in red color font.
- Section-2 presented the mathematical modelling and is sufficient. however the authors are advised to verify whether all the parameters that used in the Equations are discussed or not. It is identified that, few of the parameters are missing in the description.
The authors would like to thank to the reviewer for the valuable’s comments and suggestions. Yes, the authors agree and accept the reviewer’ comments that the authors miss few of the parameters in the description. Therefore, the manuscript has been added with the frontal area parameter in the discussion section to improve the quality of paper. Changes are marked throughout the discussion section of the revised manuscript.
- The results shown in Fig.8 are rendered by considering the driver mass up to 97Kg. The authors are advised to present the results by considering the driver mass along with the co-passenger mass (up to the weight of 150kgs).
The authors are very grateful for the comment and recommendation to improve the quality of this paper. We agree and accept the reviewer’ comment that the authors need to consider the driver mass along with the co-passenger mass up to 150 kg. Therefore, the author changed the rider mass from 70 to 150kg. Changes are marked throughout the discussion section of the revised manuscript.

Reviewer 2 Report
This manuscript proposes The effect of structure and operating parameters on the effective performance area of an electric scooter. Overall, the article is well written. Just few minor comments to improve based on reviwer stand point:
1- The "Abstract" section should be more intensively focused on the main idea directly and must contain the contribution of this manuscript with numerical result indicators.
2- The flowchart in Fig 3 should be reconsidred. The fonts should be clear and similar to the fonts used in Fig 4. It is hard to read at its current form.
3- PID controller parametres are not mentioned. Eventhough, the equation is there. Authors needs to highlight the exact parametres values.
4- Table of comparison of conducted work and related works in literature should be added to highlight the enhanced outcomes if there is any.
5- More than 50% of references are not based on last 3 years. Authors may need to update/add some new references to confirm it is uptodate. These suggsutions may be helpfull:
doi.org/10.3390/su15075866
doi.org/10.3390/en15228623
The article is well written.
Author Response
Detailed Response to Reviewers
Dear Chief Editor and Reviewers
We would like to thank you very much for your kind consideration and facilitating the review of our manuscript. I’m submitting the detailed response with consideration of all comments and recommendations given by respective editor and reviewers. I believe that their comments helped the authors to improve the quality of the manuscript to a great amount, and in this regards I appreciate the valuable time they spent on reviewing the revise manuscript. Please note that the blue colored text is our responding to the reviewers, and the red colored text is the revised/added sections to the manuscript. Also, the added sections in the manuscript are colored in red.
Le Trong Hieu
Reviewer #2
Authors reply: Thank you very much for your advice. The authors would like to response the reviewer comments below:
- The "Abstract" section should be more intensively focused on the main idea directly and must contain the contribution of this manuscript with numerical result indicators.
The authors are very grateful for the comments and suggestions to improve the quality of this paper. The author agree and accept with reviewer for the valuable comment, Therefore the authors revised a Abstract of paper by add the main idea and the contribution of manuscript with numerical results indicators.
- The flowchart in Fig 3 should be reconsidered. The fonts should be clear and similar to the fonts used in Fig 4. It is hard to read at its current form.
We would like to thank you for the valuable comment from the reviewer. Yes, we agree with the reviewer that the flowchart in Fig 3 should be rewritten. Based on this suggestion, the author made a revised flowchart. It was presented in a revised manuscript.
- PID controller parameters are not mentioned. Even though, the equation is there. Authors need to highlight the exact parameters values.
The author are very grateful for the comment and recommendation to improve the quality of this paper. Based on the suggestion’ reviewer, the authors have been made a revision in PID controller section and highlight the exact parameter.
- A table of comparison of conducted work and related works in literature should be added to highlight the enhanced outcomes if there is any.
The authors are very grateful for the comment and suggestion to improve the quality of this manuscript. Therefore, the authors have made a table of comparison of conducted work and related works in literature. It was represented in the introduction section.
- More than 50% of references are not based on last 3 years. Authors may need to update/add some new references to confirm it is up to date. These suggestions may be helpfull:
doi.org/10.3390/su15075866
doi.org/10.3390/en15228623
We would like to thank you for the valuable comment from the reviewer. Yes, we agree with the suggestion’ reviewer that the paper need to update/add some reference to confirm it is up to date. Therefore, the author added some references in the introduction section. It was presented in the revised manuscript.

Reviewer 3 Report
the manuscript has some grammatical errors and typos It is necessary to 1) better emphasise the novelty of the research in the introductory part 2) include more concepts in the introductory part related to the diffusion of electric scooters and the factors that drive people to use them either as private individuals or in sharing or renting them.
I recommend reading the following works in this regard
1) Nikiforiadis, A., Paschalidis, E., Stamatiadis, N., Raptopoulou, A., Kostareli, A., & Basbas, S. (2021). Analysis of attitudes and engagement of shared e-scooter users. Transportation research part D: transport and environment, 94, 102790.
2) Campisi, T., Basbas, S., Skoufas, A., Tesoriere, G., & Ticali, D. (2021, May). Socio-eco-friendly performance of e-scooters in Palermo: preliminary statistical results. In Innovation in Urban and Regional Planning: Proceedings of the 11th INPUT Conference-Volume 1 (pp. 643-653). Cham: Springer International Publishing.
3) Akgün-Tanbay, N., Campisi, T., Tanbay, T., Tesoriere, G., & Dissanayake, D. (2022). Modelling road user perceptions towards safety, comfort, and chaos at shared space: the via Maqueda case study, italy. Journal of advanced transportation, 2022, 1-13. We also recommend starting paragraphs with text and not with pictures Figure 3 is not readable .
I recommend increasing the size of the text in the flow chart
I recommend emphasising the limitations of the research in the concluding part
the manuscript has some grammatical errors and typos
Author Response
Detailed Response to Reviewers
Dear Chief Editor and Reviewers
We would like to thank you very much for your kind consideration and facilitating the review of our manuscript. I’m submitting the detailed response with consideration of all comments and recommendations given by respective editor and reviewers. I believe that their comments helped the authors to improve the quality of the manuscript to a great amount, and in this regards I appreciate the valuable time they spent on reviewing the revise manuscript. Please note that the blue colored text is our responding to the reviewers, and the red colored text is the revised/added sections to the manuscript. Also, the added sections in the manuscript are colored in red.
Le Trong Hieu
Reviewer #3
Authors reply: Thank you very much for your valuable comments. The authors would like to response the reviewer comments below:
- The manuscript has some grammatical errors and typos It is necessary to 1) better emphasize the novelty of the research in the introductory part 2) include more concepts in the introductory part related to the diffusion of electric scooters and the factors that drive people to use them either as private individuals or in sharing or renting them.
We would like to thank the reviewer for the valuable comments and suggestions. Yes, the authors authors accept the reviewer’s comments to improve grammatical errors, emphasize the novelty, add more concepts in the introductory part related to the diffusion of electric scooters and the factors that drive people to use them either as private individuals or in sharing or renting them. Therefore, the authors have corrected the grammatical errors in the revised manuscript. The authors have added the novelty of research in the end of introduction section. Furthermore, the authors have increased some references related to the factors that drive people to use them either as private individuals or in sharing or renting electric scooters. Changes are marked throughout the revised manuscript.
- I recommend reading the following works in this regard:
- Nikiforiadis, A., Paschalidis, E., Stamatiadis, N., Raptopoulou, A., Kostareli, A., & Basbas, S. (2021). Analysis of attitudes and engagement of shared e-scooter users. Transportation research part D: transport and environment, 94, 102790.
- Campisi, T., Basbas, S., Skoufas, A., Tesoriere, G., & Ticali, D. (2021, May). Socio-eco-friendly performance of e-scooters in Palermo: preliminary statistical results. In Innovation in Urban and Regional Planning: Proceedings of the 11th INPUT Conference-Volume 1 (pp. 643-653). Cham: Springer International Publishing.
- Akgün-Tanbay, N., Campisi, T., Tanbay, T., Tesoriere, G., & Dissanayake, D. (2022). Modelling road user perceptions towards safety, comfort, and chaos at shared space: the via Maqueda case study, italy. Journal of advanced transportation, 2022, 1-13. We also recommend starting paragraphs with text and not with pictures Figure 3 is not readable.
We would like to thank the reviewer for the valuable comments and suggestions. Based on the valuable suggestions, the authors have been made revision to the manuscript by adding the recommended reference from reviewer in introduction section to improve the quality of this manuscript. Beside that, the flowchart in Fig.3 also changed and rewritten. The changes were written in the red color font.
- I recommend increasing the size of the text in the flow chart.
The authors are very grateful for the comment and suggestions to improve the quality of this paper. The size of text flow chart was increased. Change is marked throughout the revised manuscript.
- I recommend emphasising the limitations of the research in the concluding part.
The authors are very thankful for the valuable suggestion from the reviewer. Therefore, the authors have been made a limitation of the research in the concluding part. Change is marked throughout the revised manuscript.

Round 2
Reviewer 1 Report
The paper can be accepted as the authors revised the paper as per the comments.
Author Response
Detailed Response to Reviewers
Dear Chief Editor and Reviewers
We would like to thank you very much for your kind consideration and facilitating the review of our manuscript. I’m submitting the detailed response with consideration of all comments and recommendations given by respective editor and reviewers. I believe that their comments helped the authors to improve the quality of the manuscript to a great amount, and in this regards I appreciate the valuable time they spent on reviewing the revise manuscript. Please note that the blue colored text is our responding to the reviewers, and the red colored text is the revised/added sections to the manuscript. Also, the added sections in the manuscript are colored in red.
Le Trong Hieu
Reviewer #1
Authors reply: Thank you very much for your valuable comments. The authors would like to response the reviewer comments below:
- The paper can be accepted as the authors revised the paper as per the comments.
The authors would like to thank to the reviewer for the valuable’ comment.

Reviewer 3 Report
Dear authors
the manuscript still has some grammatical errors and typos .
We also recommend that you
1) revise the list of references in accordance with the journal template .
2) revise keywords
3) insert tables in accordance with the journal template

Minor editing of English language required
Author Response
Detailed Response to Editor and Reviewers
Dear Chief Editor and Reviewers
We would like to thank you very much for your kind consideration and facilitating the review of our manuscript. I’m submitting the detailed response with consideration of all comments and recommendations given by respective editor and reviewers. I believe that their comments helped the authors to improve the quality of the manuscript to a great amount, and in this regards I appreciate the valuable time they spent on reviewing the revise manuscript. Please note that the blue colored text is our responding to the reviewers, and the red colored text is the revised/added sections to the manuscript. Also, the added sections in the manuscript are colored in red.
Le Trong Hieu
Reviewer #3
Authors reply: Thank you very much for your valuable comments. The authors would like to response the reviewer comments below:
- Revise the list of references in accordance with the journal template ..
We would like to thank the reviewer for the valuable comments and suggestions. The authors have been made revision to the manuscript by changing the list of references follow with the journal template. Changes are marked throughout the revised manuscript.
- Revise keywords.
We would like to thank the reviewer for the valuable comments and suggestions. The keywords were updated by the authors. The changes were written in the red color font.
- Insert tables in accordance with the journal template.
The authors are very grateful for the comment and suggestions to improve the quality of this paper. The table was changed following the journal template. Changes are marked throughout the revised manuscript.
- Minor editing of English language required.
The authors are very thankful for the valuable suggestion from the reviewer. With the help from native English speakers, the authors have checked and fix the shortcomings, sentences construction, typos and wrong words through the paper.
